# Lipid-Associated GWAS Loci Predict Antiatherogenic Effects of Rosuvastatin in Patients with Coronary Artery Disease

**DOI:** 10.3390/genes14061259

**Published:** 2023-06-13

**Authors:** Stanislav Kononov, Iuliia Azarova, Elena Klyosova, Marina Bykanova, Mikhail Churnosov, Maria Solodilova, Alexey Polonikov

**Affiliations:** 1Department of Internal Medicine No. 2, Kursk State Medical University, 3 Karl Marx Street, 305041 Kursk, Russia; 2Department of Biological Chemistry, Kursk State Medical University, 3 Karl Marx Street, 305041 Kursk, Russia; 3Laboratory of Biochemical Genetics and Metabolomics, Research Institute for Genetic and Molecular Epidemiology, Kursk State Medical University, 18 Yamskaya Street, 305041 Kursk, Russia; 4Department of Biology, Medical Genetics and Ecology, Kursk State Medical University, 3 Karl Marx Street, 305041 Kursk, Russia; 5Laboratory of Genomic Research, Research Institute for Genetic and Molecular Epidemiology, Kursk State Medical University, 18 Yamskaya Street, 305041 Kursk, Russia; 6Department of Medical Biological Disciplines, Belgorod State University, 85 Pobedy Street, 308015 Belgorod, Russia; 7Laboratory of Statistical Genetics and Bioinformatics, Research Institute for Genetic and Molecular Epidemiology, Kursk State Medical University, 18 Yamskaya Street, 305041 Kursk, Russia

**Keywords:** coronary artery disease, plasma lipids, carotid intima-media thickness, pharmacogenetics, personalized medicine, single nucleotide polymorphisms, lipid-lowering therapy, rosuvastatin

## Abstract

We have shown that lipid-associated loci discovered by genome-wide association studies (GWAS) have pleiotropic effects on lipid metabolism, carotid intima-media thickness (CIMT), and CAD risk. Here, we investigated the impact of lipid-associated GWAS loci on the efficacy of rosuvastatin therapy in terms of changes in plasma lipid levels and CIMT. The study comprised 116 CAD patients with hypercholesterolemia. CIMT, total cholesterol (TC), low-density lipoprotein cholesterol (LDL-C), and triglycerides (TG) were measured at baseline and after 6 and 12 months of follow-up, respectively. Genotyping of fifteen lipid-associated GWAS loci was performed by the MassArray-4 System. Linear regression analysis adjusted for sex, age, body mass index, and rosuvastatin dose was used to estimate the phenotypic effects of polymorphisms, and *p*-values were calculated through adaptive permutation tests by the PLINK software, v1.9. Over one-year rosuvastatin therapy, a decrease in CIMT was linked to rs1689800, rs4846914, rs12328675, rs55730499, rs9987289, rs11220463, rs16942887, and rs881844 polymorphisms (Pperm < 0.05). TC change was associated with rs55730499, rs11220463, and rs6065906; LDL-C change was linked to the rs55730499, rs1689800, and rs16942887 polymorphisms; and TG change was linked to polymorphisms rs838880 and rs1883025 (Pperm < 0.05). In conclusion, polymorphisms rs1689800, rs55730499, rs11220463, and rs16942887 were found to be predictive markers for multiple antiatherogenic effects of rosuvastatin in CAD patients.

## 1. Introduction

Atherosclerosis is a complex disease whose pathogenesis includes numerous pathological conditions, such as increased plasma cholesterol, its deposition in the arterial wall, endothelial dysfunction, and vascular remodeling. These processes are influenced both by environmental and genetic factors [1,2]. It is widely agreed that dyslipidemia plays a crucial role in the development of atherosclerosis and coronary artery disease (CAD) [2]. Lipid metabolism abnormalities and coronary heart disease are determined by substantial effects of genetic factors [3,4,5]. According to the GWAS Catalog (https://www.ebi.ac.uk/gwas/home, accessed on 14 February 2019), approximately 1700 genome-wide association studies (GWAS) on over 450 cardiovascular phenotypes have been conducted, yielding substantial insights into the genetic etiology of cardiovascular disease and related traits [6]. Numerous GWAS, including large-scale studies by the Global Lipids Genetics Consortium, have discovered loci associated with plasma lipids [7,8,9,10,11,12]. GWAS were also conducted to find genetic determinants of statin pharmacogenetics, including JUPITER, the largest study of the lipid-lowering effects of rosuvastatin [13,14,15]. Statins are the first line of lipid-lowering drugs widely used in both primary and secondary prevention of CAD. For one of the most popular statins, rosuvastatin, it was shown that it could reduce both blood lipid levels, including total cholesterol (TC), LDL-C (low-density lipoprotein cholesterol), triglycerides (TG), and carotid intima-media thickness (CIMT) [16].

Carotid intima-media thickness is a surrogate marker for the presence and progression of atherosclerosis and is useful for evaluating the risk and incidence of cardiovascular disease [17]. A causal relationship between LDL-C and CIMT has been observed [18], and CIMT is being considered a marker of preclinical atherosclerosis [18,19]. A growing body of evidence has been provided to consider CIMT as a predictive marker for coronary artery atherosclerosis, its severity, and the extent of plaque burden [20,21,22,23]. Importantly, rosuvastatin may reduce not only CIMT but also atherosclerotic plaque growth [24,25]. It is impossible to claim definitively whether the hypolipidemic effects of rosuvastatin are solely responsible for the regression of atherosclerosis because the pleiotropic effects of the drug may contribute to arterial wall changes through inflammation reduction [25,26]. The mechanisms by which rosuvastatin is responsible for vascular wall changes may be unraveled by pharmacogenetic studies; however, a majority of such studies have investigated only the lipid-lowering effects of the drug [13,14,27,28]. Our research team has conducted several studies on the effect of gene polymorphisms on plasma lipids and CIMT, as well as the dynamics of their changes during rosuvastatin lipid-lowering therapy [29,30,31]. It has been discovered that some lipid-associated GWAS loci contribute to the lipid-lowering effects associated with rosuvastatin therapy and determine CIMT regression. Without a doubt, CIMT seems to be a surrogate marker of atherosclerosis in CAD patients, despite the fact that it correlates with disease severity [20,21,22,23]. However, in terms of clinical application, CIMT regression during rosuvastatin therapy may mirror vascular wall changes and be indicative of drug efficacy. The aim of our pharmacogenetic study was to identify the effects of fifteen lipid-associated GWAS loci such as rs4846914 (*GALNT2*), rs11220463 (*ST3GAL4*), rs881844 (*STARD3*), rs1689800 (*ZNF648*), rs12328675 (*COBLL1*), rs9987289 (*PPP1R3B*), rs55730499 (*LPA*), rs3136441 (*F2*), rs6065906 (*PLTP*), rs838880 (*SCARB1*), rs386000 (*LILRA3*), rs1883025 (*ABCA1*), rs3764261(*CETP*), rs217406 (*NPC1L1*), and rs16942887 (*PSKH1*) on plasma lipids and CIMT in CAD patients taking rosuvastatin. These loci have recently been observed to be associated with CAD susceptibility and CIMT [32].

## 2. Materials and Methods

*Study design*. The present research was designed as a prospective study, in which the participants received rosuvastatin at an initial dose of 5 mg. Then, in 4 weeks, we controlled the levels of LDL-C. In cases where the levels were below the target level (1.8 mmol/L), patients continued taking the dose, which let them attain the target level. If it was not attained, the dose of rosuvastatin was gradually increased to 10, 20, and finally the maximum dose of 40 mg. The control of LDL-C levels was performed every 4 weeks after the dose increase. For the association analysis, we used the change (delta, Δ) in TC, LDL-C, TG, and CIMT values over the first 6 and 12 months of the study.

*Study participants*. The patient cohort comprised 116 Russian patients with the diagnosis of CAD stable angina pectoris grade II–III. The study participants were described in detail in our previous work [31]. Characteristics of the study cohort are given in Table 1. In brief, the cohort included 85 men (73%) and 31 postmenopausal women (27%), with a mean age of 61.0 ± 7.25 years. Approximately 58% of CAD patients experienced a myocardial infarction in the past, and 97.5% of study participants had arterial hypertension. The diagnosis of CAD was confirmed by qualified cardiologists according to the Canadian Cardiovascular Society’s grading of angina, ECG stress tests (treadmill test), and 24-h Holter’s ECG monitoring. TC levels higher than 4.0 mmol/L and LDL-C levels higher than 1.8 mmol/L were considered criteria for dyslipidemia.

*Biochemical and ultrasound investigations* were described in our previous paper [31]. Plasma lipid levels, except LDL-C, were detected using an automatic laboratory analyzer. LDL-C was calculated using Friedewald’s equation. CIMT measurement was performed in B-mode using the standard method at the distal third of the common carotid artery at a distance of 1–1.5 cm proximal to the bifurcation along the posterior wall [33]. The following two parameters of carotid intima-media thickness were used in the study: mean CIMT (the mean value of all measurements on the right and left sides) and maximum CIMT (the maximum value obtained from measurements on both sides with the further assessment of its change on the side where it was the highest).

*Genetic analysis*. Single nucleotide polymorphisms (SNPs) selected for this study have previously been found to be associated with plasma lipids in several genome-wide association studies [7,8,9,10,11,12] (GWAS Catalog, (https://www.ebi.ac.uk/gwas, accessed on 13 March 2023). As mentioned above, some of the SNPs have been recently found to be associated with CAD susceptibility and CIMT [32], and we were interested in investigating whether these variants determine the lipid- and CIMT-lowering effects of rosuvastatin in CAD patients. The investigated SNPs included rs4846914 (*GALNT2*), rs11220463 (*ST3GAL4*), rs881844 (*STARD3*), rs1689800 (*ZNF648*), rs12328675 (*COBLL1*), rs9987289 (*PPP1R3B*), rs55730499 (*LPA*), rs3136441 (*F2*), rs6065906 (*PLTP*), rs838880 (*SCARB1*), rs386000 (*LILRA3*), rs1883025 (*ABCA1*), rs3764261 (*CETP*), rs217406 (*NPC1L1*), and rs16942887 (*PSKH1*).

Extraction of DNA was performed from venous blood samples using the phenol-chloroform method and precipitation with ethanol. Genotyping was performed using iPLEX technology on the MassARRAY 4 system (Agena Bioscience, San Diego, CA, USA). The software MassARRAY Assay Design Suite was used to select a primer set and to design a multiplex panel for SNP genotyping. The sequences of primers are available on request.

*Statistical analysis*. The normality of the distribution of lipid and CIMT values was determined using the Kolmogorov–Smirnov and Shapiro–Wilk tests. As the trait distributions deviated from the normal one, the values were expressed as the median and interquartile range (Me, Q1; Q3). The statistical package STATISTICA v13.0 (Statsoft, Tulsa, OK, USA) was utilized for descriptive statistics, distribution analysis, and determining the significance of lipid and CIMT changes. The significance of lipid and CIMT changes during rosuvastatin therapy was tested by Wilcoxon’s matched pairs test. The distribution of genotype frequencies according to Hardy–Weinberg equilibrium was assessed with Fisher’s exact test. For the association analysis, we used linear regression with adjustments for sex, age, body mass index (calculated as body mass in kilograms divided by height in meters in a square), and rosuvastatin dose, which allowed the participants to attain the target LDL-C level. The change in lipid levels was calculated as the difference between the natural log-transformed on- and off-treatment levels divided by the natural log-transformed off-treatment level, as described by Postmus et al. [13]. We applied the same approach for the CIMT change, but before the logarithmic transformation, the CIMT data were multiplied by 10. Empirical *p*-values (Pperm) were calculated through the adaptive permutation procedure. We tested three genetic models for SNP-phenotype associations, such as additive, dominant, and recessive. The PLINK v1.92 software [34] was used for all genetic calculations, including estimation of minor allele frequencies (MAF), tests for Hardy-Weinberg equilibrium (HWE), multiple regression analysis, and permutation procedures. When the *p*-value was less than 0.05, all of the results were declared statistically significant.

## 3. Results

The majority of the patients (N = 110, 94.8%) attained the target LDL-C level during rosuvastatin therapy. The change in lipid and CIMT values was assessed during 6 and 12 months of observation. The reduction of all lipid parameters (*p* < 1 × 10^6^ for TC, LDL-C, and TG changes during 6- and 12-month periods) was significant, except HDL-C levels (*p* = 0.16 and 0.87, respectively). Reductions in the maximum (*p* < 1 × 10^6^) and mean CIMT (*p* = 0.025 and <1 × 10^5^ for the change in 6 and 12 months, respectively) were also significant (Appendix A).

### 3.1. Associations of the SNPs with Lipid and CIMT Reduction during the 6-Month Therapy by Rosuvastatin

The distribution of genotypes for the majority of SNPs (except for rs16942887) was in Hardy-Weinberg equilibrium. The observed effects of SNPs on the lipid-lowering effect and CIMT regression in CAD patients taking rosuvastatin for the 6-month period are shown in Table 2. We found that plasma TC reduction was associated with two SNPs, such as rs55730499 of *LPA* (beta = 0.323, Pperm = 0.0022, recessive effect) and rs6065906 of *PLTP* (beta = 0.140, Pperm = 0.0135, recessive effect). Based on the beta coefficients, it can be estimated that the carriage of homozygous genotypes for the minor alleles of the above polymorphisms was associated with 32.3% and 14.0% worse TC reduction, respectively, compared with the carriers of alternative genotypes (the positive beta indicates a worse lipid reduction and consequently a worse statin response). LDL-C reduction was found to be associated with rs55730499 of the *LPA* gene (Pperm = 0.0224, recessive effect) and rs1689800 of the *ZNF648* gene (Pperm = 0.0493, additive effect). The rs1689800G allele was associated with a 4.6 percent worse reduction in plasma LDL-C levels after 6 months of rosuvastatin therapy. The effect size for the rs55730499 polymorphism of *LPA* was greater in the TT homozygotes and characterized by a 50.4% worse decrease in LDL-C concentration than in carriers of alternative genotypes.

As can be seen from Table 2, four lipid-associated GWAS loci were associated with both maximum and mean CIMT change: rs4846914 in *GALNT2* (Pperm = 0.0133 and 0.0344 for the association with maximum and mean CIMT change, respectively), rs11220463 in *ST3GAL4* (Pperm = 0.0159 and 0.0243, respectively), rs16942887 in *PSKH1* (Pperm = 0.0421 and 0.0483, respectively), and rs881844 in *STARD3* (Pperm = 0.0086 and 0.0033, respectively). SNP rs1689800 in the *ZNF648* gene (Pperm = 0.0234) was associated with a change in the maximum CIMT. The most significant association with TC and LDL-C reduction during 6 months of rosuvastatin therapy was found for SNP rs55730499 in *LPA*, whereas the rs881844 polymorphism of the *STARD3* gene showed the most substantial association with the regression of CIMT in CAD patients.

Table 3 shows associations between the SNPs and plasma triglyceride reduction. It has been revealed that the only SNP rs838880 of the *SCARB1* gene was associated with TG reduction (beta = 1.736, Pperm = 0.0478, recessive effect) after 6 months of lipid-lowering therapy.

### 3.2. Associations of SNPs with Lipid and CIMT Reduction during 12-Month Therapy by Rosuvastatin

Table 4 shows the associations of studied SNPs with the 12-month lipid-lowering effect and carotid intima-media thickness change in CAD patients receiving rosuvastatin.

We observed that two polymorphisms contributed to TC reduction: rs55730499 in *LPA* (beta = 0.364, Pperm = 0.0001, recessive effect) and rs11220463 in *ST3GAL4* (beta = −0.181, Pperm = 0.0273, recessive effect) after one year of lipid-lowering therapy. For the rs11220463 polymorphism, the homozygous genotype for the minor allele T was associated with an 18.1% more pronounced TC-lowering effect than in carriers of alternative genotypes. The reduction in plasma LDL-C levels was associated with rs55730499 of *LPA* (Pperm = 0.0415, recessive effect) and rs16942887 of *PSKH1* (Pperm = 0.0175, dominant effect).

Three SNPs were associated with both maximum and mean CIMT change: rs1689800 of *ZNF648* (better statin response; Pperm = 0.0105 and 0.0282 for the association with maximum and mean CIMT change, respectively); rs12328675 of *COBLL1* (worse statin response; Pperm = 0.0213 and 0.0056, respectively), and rs9987289 of *PPP1R3B* (worse statin response; Pperm = 0.0359 and 0.0109, respectively). Two polymorphisms, such as rs55730499 of *LPA* (Pperm = 0.0146) and rs881844 of *STARD3* (Pperm = 0.0223), were associated with the mean CIMT change.

During 12 months of lipid-lowering therapy with rosuvastatin, the most significant effect on the TC reduction was found for SNP rs55730499 of *LPA*, whereas the most significant effects of rs16942887 of *PSKH1* and rs12328675 of *COBLL1* were found on the reduction of LDL-C and CIMT, respectively. Associations of TG reduction (Table 3) with genotype in a 12-month period were found for rs1883025 of the *ABCA1* gene (beta = −0.7246, Pperm = 0.0160, dominant effect) and for rs11220463 in *ST3GAL4* (beta = 3.624, Pperm = 0.05, recessive effect).

## 4. Discussion

The present study demonstrated, for the first time, the impact of lipid-associated GWAS loci on CIMT (Figure 1) and plasma levels of TC, LDL-C, and TG in CAD patients during rosuvastatin therapy (Figure 2).

In particular, we found associations between SNPs located at *ZNF648*, *LPA*, *ST3GAL4*, *PSKH1*, *GALNT2*, *COBLL1*, *PPP1R3B*, and *STARD3* genes and reductions in CIMT. Pharmacogenetic associations of the variants in *ZNF648*, *LPA*, *ST3GAL4*, *PSKH1*, *PLTP*, *SCARB1* with the change in lipid levels were also found for the first time. Thus, lipid-associated GWAS loci are associated not only with lipids and the risk of coronary artery disease [32] but are also responsible for the antiatherogenic effects of rosuvastatin after 6 and 12 months of lipid-lowering therapy in patients suffering from CAD.

Potential mechanisms by which polymorphic variants of the investigated lipid metabolism genes can impact the antiatherogenic effects of rosuvastatin therapy are of interest. The *ZNF648* gene encodes a zinc finger protein 648, which may be involved in DNA-templated transcription [35]. In the present study, the rs1689800 polymorphism was associated with CIMT change. The possible mechanism of the effect of the variant on the vascular wall may include its influence on oxidative stress (GLUL catabolizes glutamate, an amino acid required for glutathione synthesis) because such a mechanism is described for the nearest SNP, rs10911021, located in the same genomic region [36].

The *GALNT2* gene encodes N-acetylgalactosaminyltransferase 2, an enzyme involved in O-linked glycosylation of substrates, including those regulating lipid metabolisms such as apolipoprotein C-III (APOC-III), angiopoietin-related protein 3 (ANGPTL3), and phospholipid transfer protein (PLTP); in these ways, it influences HDL-C and TG metabolism [37,38]. The rs4846914G allele has been linked to atherogenic changes in plasma lipid metabolism, such as an increase in TG and a decrease in HDL-C levels [9,10]. We found no associations of this variant with changes in any plasma lipid, but this SNP was significantly associated with a better CIMT regression during a 6-month period of rosuvastatin therapy. The presence of the link with CIMT change and the absence of any association with lipid level change suggest that the pharmacogenetic effect on CIMT change might be mediated through non-lipid-related mechanisms, taking into account that SNP rs4846914 is known to be associated with endothelial function, serum levels of insulin and glucose [39], as well as hypertension [40]. In particular, the latter study demonstrated higher promoter methylation of the *GALNT2* gene, higher levels of ApoB, and lower levels of ApoA1 in hypertensives with the GG genotype [40].

The *COBLL1* gene encodes a cordon-bleu WH2 repeat protein such as 1, which has actin monomer and cadherin binding activity [41]. In the present study, the variant rs12328675 was associated with worse CIMT regression (for the carriers of the C-allele, dominant effect) during 12 months of observation. This association was the most significant among all variants studied over a 12-month period. There were no associations with lipid levels changing. The link between this polymorphism and CIMT change can be explained by the ability of the risk allele C, associated with the higher CAD risk, to form transcription factor binding sites for the factors involved in the processes of vascular inflammation regulation and angiogenesis (AP-1 (syn. JUN), SMAD2, SMAD3, SMAD4, and E2F8) [42].

The *LPA* gene encodes lipoprotein (a), which is a well-known and independent risk factor for coronary artery disease [11]. The studied rs55730499 variant is associated with Lp(a) concentrations [11], coronary artery disease risk [8], myocardial infarction risk [43], and stroke risk [44]. The T-allele is associated with higher TC levels out of therapy [43], and in our study, the TT genotype (homozygous for the minor allele) was associated with reduced drug response in terms of TC, LDL-C, and CIMT change. The association with TC change had the highest significance among all associations with TC change, and moreover, rs55730499 was the only variant studied that was associated with both TC, LDL-C, and CIMT change in plasma at the same period of observation (12 months). Based on these findings, the studied *LPA* variant can be considered the first of all studied variants to be used as a predictor for rosuvastatin therapy personalization.

The *PPP1R3B* gene encodes protein phosphatase 1 regulatory subunit 3B, a key protein in hepatic glycogen metabolism [12]. The rs9987289 variant is associated with plasma TC and LDL-C levels [9,12] but was not tested for carotid atherosclerosis traits or the pharmacogenetics of rosuvastatin. In our study, the presence of the minor A-allele was associated with a worse CIMT response to rosuvastatin. Such an effect could be explained by the association of the studied SNP with inflammatory markers, such as C-reactive protein [45], known for its influence on cardiovascular risk and atherosclerosis progression [46], and also by the link of the variant with metabolic syndrome [47] and glucose levels [48], taking into account the known role of hyperglycemia in the dysfunction of the endothelium [49,50].

The *ABCA1* gene encodes the phospholipid-transporting ATPase ABCA1, which provides the efflux of intracellular cholesterol to apolipoproteins and the formation of nascent high-density lipoproteins [51]. The minor T-allele of the rs1883025 SNP is associated in GWAS with an anti-atherogenic phenotype: lower TC [9], LDL-C [52], TG [53], and HDL-C levels [52], and with a better response to statin therapy ([54], the particular drug and lipid are not specified). In the present study, we confirmed a better response to statin therapy in terms of triglyceride reduction in carriers of the minor T-allele (the dominant effect of the SNP).

The *ST3GAL4* gene encodes CMP-N-acetylneuraminate-beta-galactosamide-alpha-2,3-sialyltransferase 4, an enzyme involved in the terminal sialylation of glycoproteins and glycolipids. A beta-galactoside alpha2-3 sialyltransferase takes part in hemostasis (sialylation of plasma von Willebrand factor) and the inflammatory process (selectin-mediated rolling and adhesion of leukocytes during extravasation) [55]. The studied SNP in this gene, rs11220463, is associated with both blood lipids (TC and LDL-C) [53] and inflammation (CRP levels) [56]. In the present study, we found pharmacogenetic associations between the studied variant and TC and CIMT reduction during therapy. The association with CIMT can possibly be explained by the involvement of the product of *ST3GAL4* in inflammatory processes, taking into account the known association of systemic inflammation (assessed by the systemic immune-inflammatory index) with CIMT [57]. For rs11220463, the association with inflammation was found in terms of CRP levels [56], and CRP is known to be associated with CIMT [58,59], but this association is not proven to be causal [58].

The *PSKH1* gene encodes serine/threonine-protein kinase H1, involved in intracellular protein trafficking and pre-mRNA processing [60]. For the rs16942887 variant, we found associations with LDL-C and CIMT changes on rosuvastatin therapy. There are no reported associations in the literature between this variation and changes in LDL-C and CIMT when taking rosuvastatin. The possible mechanism of the effect of SNP is difficult to predict because of the lack of information on the mechanisms of lipid and vascular influence of *PSKH1*.

The *STARD3* gene encodes StAR-related lipid transfer protein 3, which mediates cholesterol transport from the endoplasmic reticulum to endosomes [61]. In the literature, the rs881844 polymorphism in *STARD3* was known to be associated with plasma lipids, including TC and HDL-C [43,53]. In the present study, we did not find any associations with changes in plasma lipid levels on rosuvastatin therapy; however, this SNP showed the strongest effects on CIMT changes in both 6- and 12-month periods. The effect on the vascular wall can be explained by the involvement of *STARD3* in the regulation of cholesterol-dependent inflammation and sensitivity to proinflammatory cytokines [62].

The *PLTP* gene encodes phospholipid transfer protein, which is involved in the transfer of phospholipids and free cholesterol from LDLs and VLDLs into HDLs [63], as well as the uptake of cholesterol from peripheral cells [64]. Taking into account the function of the gene product, it was rather logical to find the influence of rs6065906 in *PLTP* on the lipid-lowering effect of rosuvastatin in terms of TC reduction in the present study, which was not reported before.

Thus, the molecular mechanisms underlying the effects of the studied loci in terms of lipid and CIMT reduction remain unknown, but taking into account the biological functions of the proteins, these genes possess pleiotropic effects on biological and pathological processes including lipid metabolism, inflammation, leukocyte adhesion to the endothelium, endothelial dysfunction, and glucose metabolism. Further experimental studies are needed to explain the functional effects of polymorphisms.

It can be noticed that the association between SNPs and the change in plasma cholesterol level and CIMT was different between 6 months and 12 months. This could be explained by two reasons. The first explanation is a weak effect of polymorphisms that reached a borderline significance level at the one-time point but were not seen at another time point. However, it could be explained according to the physiology of lipid metabolism, the function of genes, and the effects of their polymorphisms. For the variant in *LPA*, the influence on CIMT can be delayed: firstly, there is drug response from lipids (because lipoprotein (a) particles contain LDLs), and then from the vascular wall (which needs more time to respond). In the same manner, rs1689800 (*ZNF648*) initially influenced blood lipids and maximum CIMT, and in time the effect on the vessels became more significant (both mean and maximum CIMT with lower *p*-value). Some genetic polymorphisms influence CIMT change independently of lipids (e.g., loci at *COBLL1* and *STARD3*). The *COBLL1* gene is known to influence angiogenesis and the regulation of inflammation in vessels [42], but its effect on plasma lipids is unknown. The delayed effect of this SNP (CIMT change over a 12-month period) can be explained by the longer time needed for vascular changes (compared with plasma lipids) as a pleiotropic effect of rosuvastatin. The *STARD3* variant showed the vascular effect faster—in a 6-month period, and it lasted for 12-months. We believe it could be a fast response to lipid concentration changes in the cells of the vascular wall because the *STARD3* gene regulates the inflammatory response to cholesterol and cholesterol transport within membrane contacts [61] (that is why it possibly did not influence plasma lipid concentration in our study). The variant at *ST3GAL4* showed first an effect on CIMT and then on lipids. According to the function of the gene, it regulates inflammation, selectin-mediated leukocyte adhesion, and arterial calcification [55], so the effect on the vessels (due to inflammation) can be realized faster. The effect size of the association with LDL-C change in the first 6 months was rather low, which is why the association did not reach the targeted significance level.

The fifteen loci selected for the current study have been discovered in GWAS as genetic variants associated with blood lipids, the risk of cardiovascular disease (angina pectoris, myocardial infarction, heart failure, and stroke), and the effectiveness of statin therapy in terms of lipid reduction [7,8,9,10,11,12,43,54]. The present study showed that lipid-associated GWAS loci are linked to the lipid-lowering effects of rosuvastatin, and these effects were observed in the reduction of CIMT. This finding provides some insights into the mechanisms of drug action, such as: (1) only CIMT decrease was associated with genotype (for example, based on the link with *COBLL1* and *STARD3* variants and no association with lipid reduction); or (2) both CIMT decrease and lipid reduction were associated with genotype (for example, the variant in *LPA*). Thus, it can be concluded that there could be diverse mechanisms of the drug effect on the vascular wall, governed by genetic factors that are dependent or independent of lipid change. Confirmation of rosuvastatin’s effect on the vascular wall using genetic association analysis supports the clinical use of the drug in CAD patients and explains inter-individual variability in the drug effect.

*Study limitations*. Due to the non-significant HDL-C change in the therapy observed in the present study, we did not test the SNPs for associations with HDL-C change. This is a limitation of our study; however, we cannot say it is a big omission because one of the main phenotypes studied here was CIMT, which is known to have a causal relationship with LDL-C and with other factors but not with HDL-C itself [18]. Furthermore, the use of a surrogate endpoint for the vascular wall, CIMT change, was a limitation of our study. Further studies using other endpoints, such as coronary plaque characteristics, could provide a more precise genetic impact on the changes in the vascular wall in CAD patients taking statins. As for CIMT change, the effect size could not be so high since the prominent change in the vascular wall could not occur in a rather short period of time. The majority of the observed associations were significant but at a borderline level, except for the variant in the LPA gene. This is because of both the relatively moderate effects of SNPs and/or the small sample size investigated with respect to rosuvastatin pharmacogenetics. Therefore, the study’s findings should be interpreted with caution, and further investigation with a larger sample size is required to reproduce the observed associations.

## 5. Conclusions

In the present pharmacogenetic study of rosuvastatin effects in CAD patients, associations between both TC (and/or LDL-C) and CIMT changes during the same follow-up period (6 or 12 months) were observed for the polymorphisms rs55730499 of *LPA* (after 12 months of therapy) and rs1689800 of *ZNF648* (after 6 months of therapy). In general, a decrease in both atherogenic lipids and CIMT was associated with the effects of four polymorphisms: rs1689800 (LDL-C and CIMT), rs55730499 (TC, LDL-C, and CIMT), rs11220463 (TC and CIMT), and rs16942887 (LDL-C and CIMT). Half of the eight investigated lipid-associated GWAS loci, such as rs4846914, rs12328675, rs9987289, and rs881844, were linked to CIMT changes but were not associated with changes in plasma lipids during rosuvastatin therapy. The rs881844 and rs12328675 polymorphisms were most strongly associated with CIMT regression, whereas rs55730499 and rs16942887 showed associations with the reduction of plasma TC and LDL-C. Thus, the rs1689800, rs55730499, rs11220463, and rs16942887 polymorphisms were found to be predictive markers for multiple antiatherogenic effects of rosuvastatin in CAD patients. Undoubtedly, the observed associations should be replicated in independent pharmacogenetic studies with a larger number of CAD patients. The successful validation of these associations may provide novel genetic markers for personalized hypolipidemic therapy with rosuvastatin and monitoring of drug effectiveness in patients with atherogenic changes in lipid metabolism.

## Figures and Tables

**Figure 1 genes-14-01259-f001:**
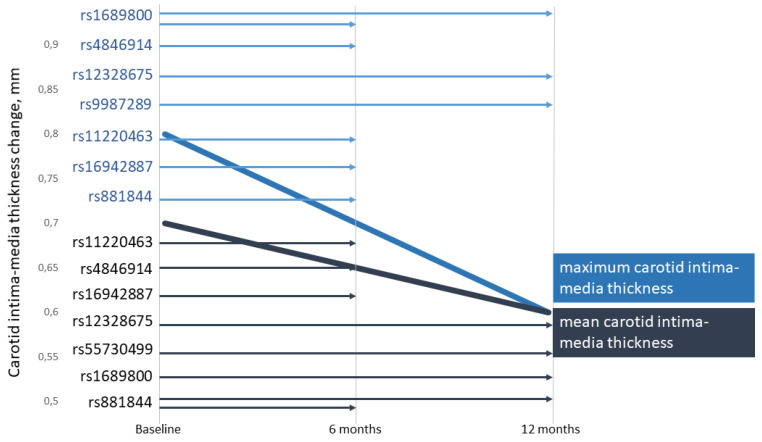
Associations of single nucleotide polymorphisms with carotid intima-media thickness (CIMT) change in coronary artery disease patients taking rosuvastatin at 6 and 12 months of therapy. The bold lines show CIMT reduction (mm), and the length of the arrows depicts the effects of particular single nucleotide polymorphisms that were observed in 6- or 12-month CIMT reduction. The color of the arrows shows the influence of the SNPs on maximum (blue) or mean (dark blue) CIMT reduction.

**Figure 2 genes-14-01259-f002:**
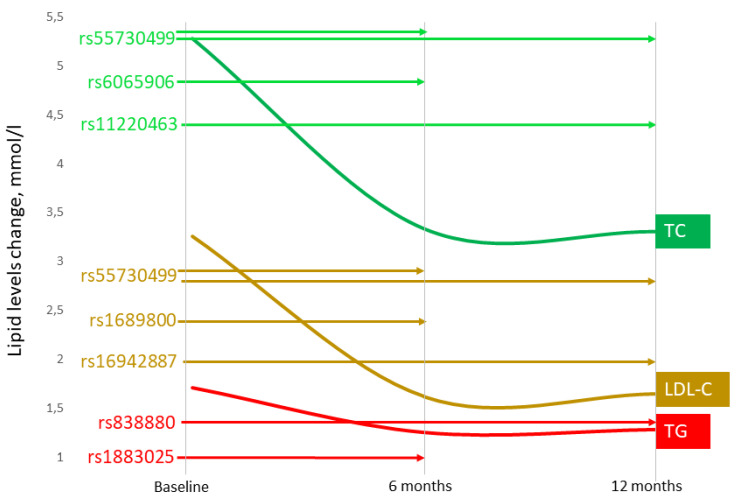
Associations of single nucleotide polymorphisms with lipid-lowering effect of rosuvastatin in coronary artery disease patients at 6 and 12 months of therapy. TC, total cholesterol; LDL-C, low-density lipoprotein cholesterol; TG, triglyceride levels. The bold lines show lipid fraction change (mmol/L), and the length of the arrows depicts the effects of single nucleotide polymorphisms, observed at 6- and 12 months. The color of the arrows shows the influence of the SNPs on TC (green), LDL-C (brown), or TG (red) reduction.

**Table 1 genes-14-01259-t001:** Baseline parameters of the study patients.

Baseline Parameter	Mean ± Standard Deviation/ Median (Q1; Q3)
Age (years)	61.0 ± 7.25
Body mass index (kg/m^2^)	28.77 ± 4.18
Hypertension (%)	97.5
Past myocardial infarction (%)	57.6
Systolic blood pressure (mmHg)	131.1 ± 8.1
Diastolic blood pressure (mmHg)	74.9 ± 4.4
Total cholesterol (mmol/L)	5.28 (4.60; 6.06)
LDL-C (mmol/L)	3.27 (2.70; 4.08)
HDL-C (mmol/L)	1.06 (0.97; 1.29)
TG (mmol/L)	1.71 (1.22; 2.37)
CIMT, maximum (mm)	0.80 (0.60; 1.00)
CIMT, mean (mm)	0.70 (0.55; 0.85)

LDL-C indicates low-density lipoprotein cholesterol; HDL-C, high-density lipoprotein cholesterol; TG, triglycerides; CIMT, carotid intima-media thickness.

**Table 2 genes-14-01259-t002:** Associations of single nucleotide polymorphisms with the 6-month lipid-lowering effect (total cholesterol, low density lipoprotein cholesterol) and carotid intima-media thickness change in coronary artery disease patients taking rosuvastatin.

Chr	Gene (SNP ID)	Effect Allele	EAF	N	Total Cholesterol		LDL-C		CIMT, Maximum		CIMT, Mean	
					Beta *	P_perm_ ^#^	Beta *	P_perm_ ^#^	Beta *	P_perm_ ^#^	Beta *	P_perm_ ^#^
1	*ZNF648* (rs1689800)	G	0.392	115	0.020	0.1786	**0.046**	**0.0493 ^A^**	**−0.084**	**0.0234 ^R^**	−0.021	0.2308
1	*GALNT2* (rs4846914)	G	0.388	115	0.004	0.7778	−0.005	0.8571	**−0.045**	**0.0133 ^A^**	**−0.038**	**0.0344 ^A^**
2	*COBLL1* (rs12328675)	C	0.170	111	−0.003	0.8571	−0.003	1.0000	0.035	0.2647	0.041	0.1000
6	*LPA* (rs55730499)	T	0.056	115	**0.323**	**0.0022 ^R^**	**0.504**	**0.0224 ^R^**	−0.010	0.8571	0.028	0.8571
7	*NPC1L1* (rs217406)	G	0.203	115	0.017	0.2308	0.033	0.3556	−0.001	1.0000	−0.003	1.0000
8	*PPP1R3B* (rs9987289)	A	0.086	115	0.019	0.3404	−0.013	0.8571	0.041	0.3404	0.051	0.1550
9	*ABCA1* (rs1883025)	T	0.263	115	0.001	1.0000	−0.004	0.8571	−0.024	0.2982	−0.025	0.2535
11	*F2* (rs3136441)	C	0.180	102	0.011	0.6429	0.025	0.7778	0.021	0.6923	0.023	0.2982
11	*ST3GAL4* (rs11220463)	T	0.190	115	−0.031	0.1280	−0.068	0.0803	**0.066**	**0.0159 ^D^**	**0.061**	**0.0243 ^A^**
12	*SCARB1* (rs838880)	C	0.336	115	0.002	1.0000	0.005	1.0000	−0.014	0.7273	−0.015	0.6923
16	*CETP* (rs3764261)	A	0.147	115	0.007	0.6250	0.027	0.8571	−0.045	0.2466	−0.042	0.1900
16	*PSKH1* (rs16942887)	A	0.116	115	0.009	0.6429	0.025	0.4643	**0.066**	**0.0421 ^D^**	**0.068**	**0.0483 ^D^**
17	*STARD3* (rs881844)	C	0.310	115	0.003	0.8571	−0.038	0.1667	**0.048**	**0.0086 ^A^**	**0.057**	**0.0033 ^A^**
19	*LILRA3* (rs386000)	C	0.203	115	0.006	0.5455	−0.001	1.0000	0.015	0.7778	0.003	0.8571
20	*PLTP* (rs6065906)	C	0.160	115	**0.140**	**0.0135 ^R^**	−0.008	1.0000	−0.022	0.6923	−0.022	0.5455

Chr, chromosome; SNP, single nucleotide polymorphism; EAF, the effect allele frequency; LDL-C, low-density lipoprotein cholesterol; CIMT, carotid intima-media thickness. Significant Pperm-values are highlighted in bold. * Beta the for difference between the natural log-transformed on- and off-treatment values adjusted for natural log-transformed off-treatment level. The beta reflects the fraction of differential lipid- or CIMT-lowering effect in carriers versus non-carriers of the minor allele (for additive model), in carriers of the genotype according to the model (dominant, recessive) versus the carriers of alternative genotypes; a negative beta indicates a better statin response (stronger lipid or CIMT reduction), a positive beta—a worse statin response. Betas were generated using linear regression analysis with age, sex, body mass index, and rosuvastatin dose as covariates. ^#^ *p*-values were generated using the permutation procedure. Superscript indicates genetic model: R, recessive; A, additive; D, dominant.

**Table 3 genes-14-01259-t003:** Associations of single nucleotide polymorphisms with triglyceride-lowering effect in coronary artery disease patients taking rosuvastatin.

Chr	Gene (SNP ID)	Effect Allele	EAF	N	6-Month Period		12-Month Period	
					Beta *	P_perm_ ^#^	Beta *	P_perm_ ^#^
1	*ZNF648* (rs1689800)	G	0.392	114	−0.6495	0.1148	−0.1946	0.4643
1	*GALNT2* (rs4846914)	G	0.388	114	0.4816	0.1919	0.05046	0.7778
2	*COBLL1* (rs12328675)	C	0.170	110	−0.2948	0.6250	−0.1431	0.7778
6	*LPA* (rs55730499)	T	0.056	114	−0.5923	0.4242	0.1825	0.6923
7	*NPC1L1* (rs217406)	G	0.203	114	0.4106	0.4118	−0.1208	0.6429
8	*PPP1R3B* (rs9987289)	A	0.086	114	−0.9197	0.1887	−0.7977	0.0756
9	*ABCA1* (rs1883025)	T	0.263	114	−0.177	0.5789	**−0.7246**	**0.0160** ** ^ D^ **
11	*F2* (rs3136441)	C	0.180	101	−0.185	0.5789	0.02512	1.0000
11	*ST3GAL4* (rs11220463)	T	0.190	114	−0.774	0.1587	**3.624**	**0.0503** ** ^ R^ **
12	*SCARB1* (rs838880)	C	0.336	114	**1.736**	**0.0478** ** ^ R^ **	0.1063	0.6429
16	*CETP* (rs3764261)	A	0.147	114	0.695	0.2931	0.02338	1.0000
16	*PSKH1* (rs16942887)	A	0.116	114	−0.6306	0.1439	−0.2626	0.3947
17	*STARD3* (rs881844)	C	0.310	114	0.4075	0.4815	−0.2337	0.2647
19	*LILRA3* (rs386000)	C	0.203	114	−0.1847	0.8571	−0.1448	0.6923
20	*PLTP* (rs6065906)	C	0.160	114	−0.4654	0.5789	−0.0718	0.8571

Chr, chromosome; SNP, single nucleotide polymorphism; EAF, effect allele frequency. Significant Pperm-values are highlighted in bold. * Beta for the difference between the natural log-transformed on- and off-treatment values adjusted for natural log-transformed off-treatment level. Betas were generated using linear regression analysis with age, sex, body mass index, and rosuvastatin dose as covariates. ^#^ *p*-values were generated using the permutation procedure. Superscript indicates genetic model: R, recessive; D, dominant.

**Table 4 genes-14-01259-t004:** Associations of single nucleotide polymorphisms with the 12-month lipid-lowering effect and carotid intima-media thickness change in coronary artery disease patients taking rosuvastatin.

Chr	Gene (SNP ID)	Effect Allele	EAF	N	Total Cholesterol		LDL-C		CIMT, Maximum		CIMT, Mean	
					Beta *	P_perm_ ^#^	Beta *	P_perm_ ^#^	Beta *	P_perm_ ^#^	Beta *	P_perm_ ^#^
1	*ZNF648* (rs1689800)	G	0.392	113	0.010	0.3091	0.024	0.2043	**−0.093**	**0.0105 ^R^**	**−0.036**	**0.0282** ** ^ A^ **
1	*GALNT2* (rs4846914)	G	0.388	113	−0.004	0.7273	−0.016	0.4118	0.002	0.8571	−0.002	0.8571
2	*COBLL1* (rs12328675)	C	0.170	109	−0.011	0.4643	−0.018	0.4643	**0.059**	**0.0213 ^A^**	**0.075**	**0.0056 ^D^**
6	*LPA* (rs55730499)	T	0.056	113	**0.364**	**0.0001 ^R^**	**0.367**	**0.0415 ^R^**	0.018	0.8571	**0.414**	**0.0146 ^R^**
7	*NPC1L1* (rs217406)	G	0.203	113	−0.002	1.0000	0.003	0.2043	0.019	0.3636	0.018	0.4516
8	*PPP1R3B* (rs9987289)	A	0.086	113	0.004	0.8571	−0.068	0.4118	**0.072**	**0.0359 ^D^**	**0.079**	**0.0109 ^D^**
9	*ABCA1* (rs1883025)	T	0.263	113	0.010	0.3148	0.001	0.4643	0.011	0.8571	0.019	0.2687
11	*F2* (rs3136441)	C	0.180	100	0.013	0.3478	0.018	0.2043	−0.011	0.6429	−0.005	1.0000
11	*ST3GAL4* (rs11220463)	T	0.190	113	**−0.181**	**0.0273** ** ^ R^ **	−0.027	0.4118	0.001	1.0000	0.032	0.1709
12	*SCARB1* (rs838880)	C	0.336	113	−0.001	1.0000	−0.005	0.4643	−0.003	1.0000	−0.004	1.0000
16	*CETP* (rs3764261)	A	0.147	113	0.017	0.3478	0.021	0.2043	−0.053	0.0833	−0.043	0.0880
16	*PSKH1* (rs16942887)	A	0.116	113	0.006	0.5789	**0.086**	**0.0175 ^D^**	−0.011	0.5217	−0.004	0.7778
17	*STARD3* (rs881844)	C	0.310	113	0.009	0.4375	0.010	0.8571	0.028	0.1852	**0.040**	**0.0223 ^A^**
19	*LILRA3* (rs386000)	C	0.203	113	0.019	0.2400	0.015	0.5200	0.026	0.2982	0.010	0.7778
20	*PLTP* (rs6065906)	C	0.160	113	0.013	0.4815	0.031	0.2571	−0.029	0.3478	−0.006	1.0000

Chr, chromosome; SNP, single nucleotide polymorphism; EAF, the effect allele frequency; LDL-C, low-density lipoprotein cholesterol; CIMT, carotid intima-media thickness. Significant Pperm-values are highlighted in bold. * Beta for the difference between the natural log-transformed on- and off-treatment values adjusted for natural log-transformed off-treatment level. The beta reflects the fraction of differential lipid- or CIMT-lowering effect in carriers versus non-carriers of the minor allele (for additive model), in carriers of the genotype according to the model (dominant, recessive) versus the carriers of alternative genotypes; a negative beta indicates a better statin response (stronger lipid or CIMT reduction), a positive beta—a worse statin response. Betas were generated using linear regression analysis with age, sex, body mass index, and rosuvastatin dose as covariates. ^#^ *p*-values were generated using the permutation procedure. Superscript indicates genetic model: R, recessive; A, additive; D, dominant.

## Data Availability

Data supporting reported results are available upon request.

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
