# Peer review of "Lipid-Associated GWAS Loci Predict Antiatherogenic Effects of Rosuvastatin in Patients with Coronary Artery Disease"

_genes, 2023, doi:10.3390/genes14061259_

Round 1

Reviewer 1 Report

       The authors investigated the contributions of some lipid-associated GWAS loci on the efficacy of rosuvastatin therapy in terms of changes in plasma lipid levels and CIMT. The results obtained by the authors can potentially help in the choice of lipid-lowering therapy. I don't have major comments.

Minor comments:

Figures 1 and 2 require a better description of the legend. It is not entirely clear what meaning the authors put into them.

It is recommended to add a "study limitations" section.

Author Response

Reviewer1

Comments and Suggestions for Authors

The authors investigated the contributions of some lipid-associated GWAS loci on the efficacy of rosuvastatin therapy in terms of changes in plasma lipid levels and CIMT. The results obtained by the authors can potentially help in the choice of lipid-lowering therapy. I don't have major comments.

  1. Figures 1 and 2 require a better description of the legend. It is not entirely clear what meaning the authors put into them.

Thank you for this suggestion. A detailed description was added to the legends.

  1. It is recommended to add a "study limitations" section.

Thank you for this comment. We discussed the limitations of our study.

Thank you so much for your generally positive assessment of our manuscript and your valuable comments and suggestions! The corrections are highlighted in yellow in the revised manuscript.

Reviewer 2 Report

1. Please supplement the general information of the"Study participants", especially the concomitant conditions related to dyslipidemia.

2. Please add relevant information about CAD patients in the conclusion and discussion section of "Study participants"

3. Please improve the format and color matching of the table, so that readers can read and understand it.

4. The analysis of the results of this paper in the "discussion" section is not complete enough, such as the status quo and prospect of relevant studies, please add. 

  • Some sentences are long and difficult to understand. Please improve them.

Author Response

Reviewer2

Comments and Suggestions for Authors

Thank you so much for generally positive assessment of our manuscript, valuable comments and suggestions!

  1. Please supplement the general information of the "Study participants", especially the concomitant conditions related to dyslipidemia.

Thank you for this comment. The information on the study participants was added to the manuscript (table with baseline characteristics).

  1. Please add relevant information about CAD patients in the conclusion and discussion section of "Study participants".

Thank you for this comment. We specified the fact that patients had CAD in the chapters you mentioned.

  1. Please improve the format and color matching of the table, so that readers can read and understand it.

Thank you for this comment. We removed the grayed-out font and left only the significance levels in bold.

  1. The analysis of the results of this paper in the "discussion" section is not complete enough, such as the status quo and prospect of relevant studies, please add.

Thank you for this comment. We provided the “Discussion” section by relevant information on the previously conducted studies. The corrections are highlighted in yellow throughout the revised manuscript.

Comments on the Quality of English Language

Some sentences are long and difficult to understand. Please improve them.

Thank you for this comment. We used English proofreading service to improve the clarity of the manuscript. Thank you again for your valuable comments and recommendations to our manuscript!

Reviewer 3 Report

To the authors

The authors analyzed the association between prespecified SNP loci and reduction of cholesterol levels and CIMT in patients receiving rosuvastatin for coronary artery disease.

The manuscript was well-written; however, I have some concerns about the methodology taken.

Major:

Methods:

1.     Some of SNP loci analyzed were in the same chromosome. Are they independent each other? This would change the way of statistical analysis.

2.     Even each SNP locus is independent between each other, there remain a problem of multiple testing. Since some of p values mentioned as statistically significant were near to 0.05, I recommend the authors to conduct appropriate statistical analysis to adjust this problem.

Discussion:

3.     It is quite interesting that the association between each locus and the change in plasma cholesterol level and CIMT was different between 6 months and 12 months.

I assume this may comes from the problem of multiple testing. Please discuss this issue and take adequate statistical means.

Or should the authors have pharmacological or genetical rationale for this difference, please clarify.

4.     Majority of loci presented as having association with change in cholesterol level or CIMT are associated with only one of these two endpoints. Is this an expression of pleiotropic effect of rosuvastatin? Or just a statistical error from multiple testing? Please discuss.

English grammar was appropriate and does not need extensive editing.

Author Response

Reviewer3

Comments and Suggestions for Authors

The authors analyzed the association between prespecified SNP loci and reduction of cholesterol levels and CIMT in patients receiving rosuvastatin for coronary artery disease.

The manuscript was well-written; however, I have some concerns about the methodology taken.

Thank you so much for generally positive assessment of our manuscript, valuable comments and suggestions!

  1. Some of SNP loci analyzed were in the same chromosome. Are they independent each other? This would change the way of statistical analysis.

Thank you for this comment. The SNPs within chromosomes 1 (rs1689800, rs4846914; R2= 0.00219175, D' = 0.0731993), 11 (rs3136441, rs11220463; R2 = 0.00997651, D' = 0.103159), and 16 (rs3764261, rs16942887; R2 = 0.00100709, D' = 0.211019) are not in linked to each other. Thus, their effects on the phenotypes are independent.

  1. Even each SNP locus is independent between each other, there remain a problem of multiple testing. Since some of p values mentioned as statistically significant were near to 0.05, I recommend the authors to conduct appropriate statistical analysis to adjust this problem.

Thank you for this comment. The tables report empirical P-values obtained by permutation tests using the PLINK software. Permutation procedure, also known as a re-sampling, is one of statistical tools solving the issue of multiple testing. In particular, PLINK provides a framework for correction for multiple testing (please, see the tutorial, first paragraph: https://zzz.bwh.harvard.edu/plink/perm.shtml), such procedure can be used for correction for multiple testing. In other words, Pperm-values can be considered as “adjusted P-values”

  1. It is quite interesting that the association between each locus and the change in plasma cholesterol level and CIMT was different between 6 months and 12 months. I assume this may come from the problem of multiple testing. Please discuss this issue and take adequate statistical means. Or should the authors have pharmacological or genetical rationale for this difference, please clarify.

Thank you for this comment. The issue of multiple testing is discussed in our previous response. All estimations of the effects we observed are moderate. This is because a sample size is relatively small to obtain more reliable estimates. We discussed this in the study limitations. We also suppose that the difference in the effects between 6 and 12 months could be explained physiologically according to the functions of gene products - we also discussed it in the "Discussion" section of the revised manuscript.

  1. Majority of loci presented as having association with change in cholesterol level or CIMT are associated with only one of these two endpoints. Is this an expression of pleiotropic effect of rosuvastatin? Or just a statistical error from multiple testing? Please discuss.

Thank you for this comment. Perhaps this finding could be explained by the effects of the studied genetic polymorphisms. We suggest that the CIMT response to rosuvastatin can be faster (for the variants influencing vessels or lipid transport within the cells) or slower (for the variants influencing plasma lipids, so more time is needed to produce changes in the vessels). The discussion of this issue was added to the "Discussion" section of the manuscript.

Comments on the Quality of English Language

English grammar was appropriate and does not need extensive editing.

Thank you again for your valuable comments and recommendations to our manuscript! We are hopeful that we have responded to your comments satisfactorily and this manuscript is now acceptable for publication.

Round 2

Reviewer 3 Report

I appreciate the revision.